# Height development and multiple bone health indicators in children aged 2–12 years with Duchenne muscular dystrophy (DMD)

**Bingying Wang**[1], **Linyuhan Zhou**[1], **Shuangru Li**[1], **Huayan Xu**[2], **Yingkun Guo**[2], **Qin Hu**[1], **Min Huang**[1], **Dan Zhou**[1], **Xiaotang Cai**[1]*, **Qiu Wang**[1]*, **Xiaomei Sun**[3]*

**1** Department of Rehabilitation Medicine, China Key Laboratory of Birth Defects and Related Diseases of Women and Children, Ministry of Education, West China Second University Hospital, Sichuan University, Chengdu, Sichuan, China, **2** Department of Radiology, China Key Laboratory of Birth Defects and Related Diseases of Women and Children, Ministry of Education, West China Second University Hospital, Sichuan University, Chengdu, Sichuan, China, **3** Department of Pediatrics, China Key Laboratory of Birth Defects and Related Diseases of Women and Children, Ministry of Education, West China Second University Hospital, Sichuan University, Chengdu, Sichuan, China

☯ These authors contributed equally to this work.
* cxt_1999@126.com (XC); 25099988@qq.com (QW); sunxiaomei@scu.edu.cn (XS)

## Abstract

### Introduction

Short stature is a frequent complication of DMD, and its pathomechanisms and influencing factors are specific to this disease and the idiosyncratic treatment for DMD.

### Purpose

To establish the height growth curve of early DMD, and evaluate the potential influencing markers on height growth, provide further evidence for pathological mechanism, height growth management and bone health in DMD.

### Methods

A retrospective, cross-sectional study of 348 participants with DMD aged 2–12 years was conducted at West China Second Hospital of Sichuan University from January 2023 to October 2023.

### Results

The growth curve for 2–12 years old boys with DMD indicates a slower growth rate compared to the average population. At age two, children with DMD have a similar height to their peers, but gradually falls behind afterwards. Short stature was observed in children with DMD before and after GC exposure, and prolonged GC use exacerbated the retardation. BMI ($\beta$ = -0.47, p = 0.007), BMD ($\beta$ = -0.005, p = 0.014), $\beta$-CTX ($\beta$ = 0.001, p = 0.002), delayed BA ($\beta$ = 0.417, p < .001), GC duration ($\beta$ = -0.006, p = 0.047) were independent influencing factors of height. Relevant bone health markers showed different sequential changing patterns.

**Data Availability Statement:** Data cannot be shared publicly because of the clinical ethical approval. Data are available from corresponding author(contact via E-mail: cxt_1999@126.com) for

researchers who meet the criteria for access to confidential data. Data are also available from the ethics committee of West China Second University Hospital (contact via Email:hx2llwyh@163.com).

**Funding:** This study was supported by grants from the Sichuan Science and Technology Support Program (2023YFG0284, 2023ZYD0121), Natural Science Foundation of Sichuan Province (24NSFC1085), Chengdu Municipal Science and Technology (2024-YF05-00493-SN) and National Natural Science Foundation of China (82271981). There were no any sponsors or funders that played any role in this study, and no additional external funding was received for this study.

**Competing interests:** The authors have declared that no competing interests exist.

**Abbreviations:** DMD, Duchenne muscular dystrophy; GC, Glucocorticoid; MDT, Multidisciplinary Team; BMI, Body mass index; BMD, Bone mineral density; BA, Bone age; CA, Chronological age; QCT, Quantitative computed tomography; β-CTX, β-isomerized C-terminal telopeptides; P1NP, Procollagen type 1 aminoterminal propeptide; N-MID, N-terminal mid-fragment of osteocalcin.

## Conclusion

The high proportion and progression of short stature are associated with the broad bone health status. Different bone indicators have different sensitivities and specificities and need to be considered together for clinical monitoring of bone health. This study provides evidence for the early monitoring of height development and relevant factors as part of bone health management in DMD, to minimize the occurrence of bone-related complications later in life.

## 1 Introduction

Duchenne Muscular Dystrophy (DMD) is a genetic disorder characterized by progressive muscle weakness and degeneration, with an estimated incidence of 1 in 3500 male live births [1]. As the disease progresses, muscles become less functional, and lose independent ambulation by a median age of 12 years old [2]. It is a well-established fact that boys with DMD often experience stunted growth and short stature compared to their healthy peers [3,4], which seems to increase emotional distress and contribute to a reduced quality of life. Glucocorticoids (GC) is the only pharmacy treatment proven to be an effective intervention for DMD patients, which can maintain muscle strength, stabilize cardio and respiratory function, and prolong ambulation by 2–5 years [5]. However, GC often leads to further deterioration of growth rate and bone health status.

To date, the etiology of short stature in DMD patients remains uncertain. Previous researchers have studied the growth patterns of DMD patients in multiple aspects. Several studies explored how genetic types influenced the growth of DMD patients, Sarrazin found more distal mutations suggest short stature, especially ones altering *Dp71* expression [6]. Another research indicated that genotype and dystrophin isoform expression influenced height trajectories, DMD participants with mutations affecting expression of *Dp71* were shorter than participants with more proximal pathogenic variants [7]. Lamb et al. researched the relationship between height and GC treatment, the earlier initiation of GC use, daily dose, longer duration of GC treatment, and higher dosage were confirmed as associated factors of shorter height [3]. The primary detrimental effect of glucocorticoid exposure on growth appears to be growth plate toxicity. GC is suspected to impair the activity of growth plates by inhibiting the differentiation of chondrocytes and osteoblasts [8], induce chondrocyte and osteoblast apoptosis [9], and disrupt the local production of paracine hormones, including insulin-like growth factor 1 (IGF-1) and C-type natriuretic peptide [10,11].

Overall, previous studies have tended to focus on genotype, GC, and other symptomatic treatments for short stature at puberty. There has been limited research on growth and development at younger ages before puberty, and there is a lack of large sample data to support this. The exact independent factors and the extent of their influence are also not clear. This study aims to establish the curve of stature development in children with early DMD in a larger sample cross-section design, and evaluate the potential influencing markers on height growth patterns, providing further evidence for pathological mechanism and clinical management of stature growth and bone health in DMD.

## 2 Methods

### 2.1 Setting and participants

This study is a retrospective, single-center, cross-sectional study. Participants who visited the pediatric rehabilitation department of West China Second University Hospital between

January 2023 and October 2023 were included, which was accessed on 25th October 2023 for research purpose, and all data were fully anonymized during and after the reseach. This study was approved by the ethics committee of West China Second University Hospital (registration number: 21PJ048), which waived the requirement for informed consent.

Inclusion criteria were as follows: (1) patients were genetically diagnosed or had muscle biopsies that confirmed DMD; (2) male; (3) aged from 2 to 12 years.

Exclusion criteria were as follows: (1) patients with other neuromuscular diseases or neuro-developmental disorders; (2) combining other pathologic causes of short stature; (3) parents with short stature.

## 2.2 Data collection and measurements

The medical records of eligible patients were reviewed to extract data. The participants were followed up in our center every 6 to 12 months, regular monitoring was conducted at least once a year, which was customized for each participant depending on their bone health condition, growth pattern, and symptoms. This included medical prescriptions, consultation notes, investigation results, scans, anthropometric data, blood biochemical indicators, and Multidisciplinary Team (MDT) correspondence between clinicians. The initial dose of GC was usually given in the form of prednisone at 0.5–0.75 mg/kg/d. At the same time as taking GC, the children were given oral vitamin D3 (800–1200 IU/d) and elemental calcium (400 mg/d) as supplements. The results of laboratory examinations were recorded, including serum levels of 25 (OH)-vitamin D, calcium, phosphorus, and plasma level of procollagen type 1 aminoterminal propeptide (P1NP), β-isomerized C-terminal telopeptides (β-CTX), N-terminal mid-fragment of osteocalcin (N-MID).

A calibrated device was used to measure height and weight, each participant was measured three times, and the average value was recorded for accuracy. For participants with scoliosis, contractures, or inability to stand independently, the ulna conversion formula was used to calculate height [12], as represented in the followinig equation: Height(cm) = (4.605* Ulna Length)+(1.308*Age in years)+28.003. Body mass index (BMI) was calculated as weight(kg)/height($m^2$). We converted the data of height to standard deviation score (Ht SDS) and z-score based on age and sex [13–15]. The vertebral bone (lumbar spine L1-L3) mineral density (BMD) was measured by a Neusoft 128-slice helical CT scanner (Neu-Viz128, China) (120 kV, 70 mAs, 3-mm slice thickness) [16]. The BMD data converted to Z-values based on age and sex were provided by the manufacturer of the QCT software (Mindways Software) [17,18]. The bone age (BA) was evaluated using the Greulich & Pyle method (TW2) [19] and scanned with the Digital Medical X-Ray Radiography System uDR 780i (Shanghai United Imaging Medical Technology Co.). Motor function was evaluated using the North Star scale. North Star Ambulatory Assessment score was based on a structured physiotherapy assessment at a pediatric rehabilitation outpatient clinic. The North Star Ambulatory Assessment score reflects their functional motor abilities, scaled from 0 (unable) to 34 [20].

## 2.3 Statistical analysis

Statistical analysis was performed using the IBM SPSS version 26.0 software. Descriptive data were expressed in mean with standard deviation (SD) and/or median values with IQR according to data types. The normality of data was determined by Shapiro-Wilk tests and the Q-Q plot. Continuous variables were analyzed using linear regression and multiple linear regression. T-test was used to measure the differences between subgroups.

## 3 Results

### 3.1 Demography and characteristics

From January 2023 to October 2023, 348 participants with DMD aged 2–12 were included. The average age was 7.97±2.51 years. 276 participants were on GC treatment, accounting for 79.3% of all participants. The mean age of GC initiation was 5.57±1.80 years and the average duration of GC use was 34.44±22.43 months. Table 1 shows the demographic characteristics of the studied population. 291 patients had at least one bone mineral density (BMD) measurement, with a mean BMD of 133.70±30.91 mg/cm$^3$. The rate of low BMD (BMD Z-score<-2) in this cohort was 31.3%.

### 3.2 Growth pattern of DMD

Fig 1A showed the proportion of different height standard deviation scores (Ht SDS) values changed by chronological age (CA). Normal stature was defined as a height above -2 standard

**Table 1. Clinic characteristics of participants.**

| Variables | N (%)or mean (±SD) |
|---|---|
| Age(years) | 7.97 (±2.51) |
| Height (cm) | 117.92 (±12.96) |
| SDS>-1 | 92(26.4%) |
| SDS -1~-2 | 122(35.1%) |
| SDS<-2 | 134(38.5%) |
| Weight (kg) | 25.38 (±8.44) |
| BMI (kg/m$^2$) | 17.84 (±3.40) |
| Normal | 222 (63.7%) |
| Overweight | 59 (17%) |
| Obesity | 67 (19.3%) |
| Vertebral fractures | 0(0%) |
| Non-ambulation | 30(8.6%) |
| Glucocorticoids | |
| Yes | 275(79%) |
| No | 73(21%) |
| Age of GC initiation (years) | 5.57 (±1.80) |
| GC use duration (months) | 34.44 (±22.43) |
| <1y | 46 (13.2%) |
| 1-2y | 61 (17.5%) |
| 2-3y | 48 (13.8%) |
| 3-4y | 46 (13.2%) |
| 4-5y | 34 (9.8%) |
| >5y | 40 (11.5%) |
| Bone mineral density (mg/cc) | 133.70 (±30.91) |
| BMD Z-score | -1.53 (±1.05) |
| >-2 | 200(68.7%) |
| <-2 | 91(31.3%) |
| BA (percentile) | 7.31(±2.52) |
| >75th | 34 (9.8%) |
| 25th-75th | 183 (52.6%) |
| <25th | 131 (37.6%) |

†Abbreviations: BMI = body mass index; GC = Glucocorticoids; BMD = bone mineral density; BA = bone age.

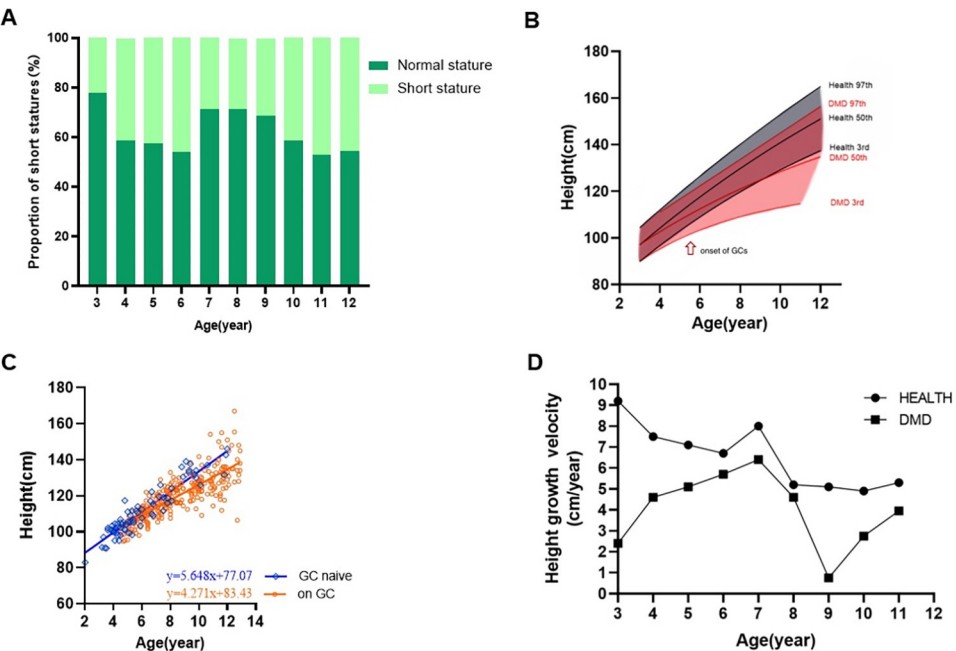

**Fig 1. Changes in height growth with age.** (A. proportion of normal and short stature changed with age (normal stature: (normal stature: Ht SDS >-2, short stature: Ht SDS <-2)), B. height growth curve of healthy males and DMD group, C. age and height scatterplot of GC naive and on GC therapy participants with DMD, D. height growth rate of healthy population and DMD group).

deviation (SD), while short stature was a height below -2SD [21]. A tendency for proportion of normal height to decrease with age can be observed.

As shown in Fig 1B, the 3rd, 50th, and 97th percentile of the height growth curve for DMD individuals showed a retarded growth compared with the healthy male population aged 2–12 years. Children with DMD were around the same height as average normal children at the age of two, but the height began to gradually fall behind after that. Totally 38.5% of participants in our cohort matched the diagnostic criteria for short stature (Ht SDS<-2), 23.9% of participants without taking GC, and 42.5% of participants receiving GC therapy were short stature. Scatters in Fig 1C show the age and height scatterplot of GC naive and on GC therapy participants with DMD, and linear regression equations of the two groups of participants. Fig 1D shows the height growth rate of the healthy male population and DMD participants in our cohort. During 3 to 11 years old, the height growth rate of DMD participants was always slower than healthy male population. The growth velocity of DMD children showed a slow climbing tendency before the age 7.

We also corrected Ht SDS by BA (instead of CA), where the incidence rates of short stature dropped from 38.5% to 20.4%, and 36.5% of individuals with short stature were still short after the Ht SDS was corrected by BA.

## 3.3 Factors affecting height and short stature

305 of 342 participants (89.2%) had normal serum levels of phosphorus (1.45–2.1nmol/L), 324 of 343 participants (94.5%) had normal serum levels of calcium (2.25–2.67nmol/L), 121 of 342 participants (35.4%) below normal serum level of 25(OH)-D (20ng/mL or 50nmol/L). Individuals in our study were under standard treatment, 25(OH)-D, calcium treatment, and regular follow-up monitoring have been carried out, therefore, the results show calcium, phosphorus,

**Table 2. Multiple linear regression analysis between height z-score and relevant factors.**

|  | r | p-value |
|---|---|---|
| ANOVA | 0.648 | <0.001 |
| Variable | β | p-value |
| Northstar Ambulatory Assessment | -0.005 | 0.339 |
| BMI* | -0.47 | 0.007 |
| 25(OH)-D | 0.002 | 0.713 |
| Phosphorus serum | -0.512 | 0.170 |
| Plasma P1NP | 0.000 | 0.788 |
| Plasma β-CTX* | 0.001 | 0.002 |
| BMD* | -0.005 | 0.014 |
| Duration of GC use* | -0.006 | 0.047 |
| BA delay* | 0.417 | <0.001 |

*The results were statistically significant.

†Abbreviations: BMI = body mass index; BMD = bone mineral density; β-CTX = β-isomerized C-terminal telopeptides; P1NP = procollagen type 1 aminoterminal propeptide; 25(OH)-D = 25-hydroxyvitamin D.

BA delay: The difference between CA and BA.

and 25(OH)-D were not associated with height of DMD participants. Candidate variables with a p-value <0.2 on univariate analysis were included in multivariable analysis. The linear regression analysis results show that BMI (β = -0.47, p = 0.007), BMD (β = -0.005, p = 0.014), β-CTX (β = 0.001, p = 0.002), duration of GC use (β = -0.006, p = 0.047), the difference between CA and BA (β = 0.417, p<0.001) were independent factors of height z-score (r = 0.648, p < .001) (Table 2).

## 3.4 Data changes over time of GC exposure

The bar graph (Fig 2) demonstrates the changing prevalence of short height (Ht SDS <-2) and low BMD (Z-score <-2), delayed BA (<25th percentile) by the duration of GC use. The trend plots indicate different sequential changing patterns in relevant bone health markers. Within one year of GC initiation, β-CTX levels decreased significantly, while the proportion of delayed BA continued to increase. In contrast, the proportion of low BMD increased only after three years of prolonged GC use. These changes in bone health markers were associated with an increasing severity in short stature. We examined the effect of GC exposure duration on proportion of short stature (OR = 0.462, 95%CI [1.056, 1.309], p<0.001), delayed BA (OR = 0.369, 95%CI [1.105, 1.375], p<0.001) and low BMD (OR = 0.127, 95%CI [1.340, 1.793], p<0.001) by binary logistic regression, and on level of β-CTX (R = 0.417, p<0.001) by linear regression.

Chi-square analyses were applied to further study the changes in GC duration. Generally, the percentage of short stature significantly grew from 24.7% to 41.3% within the first year of taking GC (p = 0.004). Also, the percentage of delayed BA (<25th percentile) was 26.6% of the GC naïve group, which significantly increased to 44.1% after GC exposure duration longer than 5 years (p = 0.002). On the contrary, the percentage of low BMD (Z score <-2) kept declining during the first two years of GC use but started to increase after two years of taking GC, from 10.3% to 44.9% (p<0.001), where a delayed deterioration was identified.

Changes in bone marker levels were studied. The median serum level of β-CTX dropped from 1306 pg/mL to 880.1 pg/mL within the first year (p<0.001). T-test analyses showed no significant difference in β-CTX between short-stature individuals and normal-stature individuals before the use of GC (p = 0.494). However, after receiving GC therapy, the β-CTX of

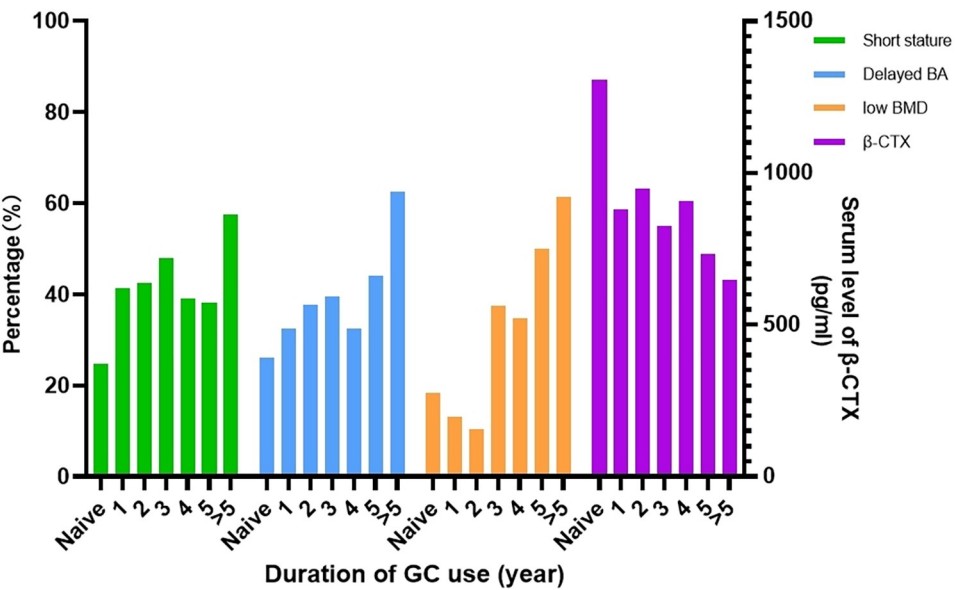

**Fig 2. Changes in bone health-related factors with duration of GC use.**

short-stature participants was significantly lower than that of normal-stature participants (p = 0.023).

## 4 Discussion

Short stature is a frequent complication of DMD, and its pathomechanisms are specific to this disease, which is not identical to other common pathogenetic causes to short stature. Previous studies have shown that the dosage, types, and exposure duration of GC use [3], as well as mutations in the distal part of the dystrophin gene [4,6], are all associated with short stature. Meanwhile, long-term use of GC has been shown to exacerbate the short stature in DMD in multiple dimensions. However, the correlation between bone health and height at different ages and their associated influencing factors remained unclear. To obtain more information about growth patterns and potential impact factors of bone health, we explored the height development in DMD participants and investigated the correlation between height and various clinical indices.

Our study showed that children aged 2–12 years with DMD tend to have similar height to healthy males at early stages of the disease. However, as the disease progressed, the height growth curve gradually deviated from the normal growth curve even before GC initiation, and prolonged exposure of GC exacerbate height growth retardation, which indicates a natural stature progression oriented from disease itself. This is consistent with the conclusion of the previous study of 34 DMD patients that height/length was normal at birth, but a gradual slow-down of height growth was observed during their early years [22]. In our cohort, 23.9% of participants who had never taken GC were short, which is higher than the third percentile of healthy males, while 42.5% of boys with DMD on GC use were short. Even when we compared bone height percentiles corrected for age by BA, there were still 20.4% of participants with short stature. It was consistent with 25% of boys with DMD being short prior to initiation of GC [23] and 45% of DMD boys on GC were short stature [6]. The results suggested that the disease leads to short stature at an earlier age, as early as around 2–5 years of age, boys with DMD are already showing deviations from the growth curve, and GC exposure exacerbates

the growth lag. The height growth velocity of the aged 3–7 DMD group was retarded to healthy males of the same age, although showed a slow climbing tendency which is opposite to healthy males.

Furthermore, to explore potential factors impacting the growth and development of boys with DMD, we analyzed the duration of GC use, BMD, BA, electrolytes (Calcium, Phosphorus, 25(OH)-D), motor ability scores (NSAA), and bone turnover markers (PINP, β-CTX, N-MID). The results are quite valuable, showing that BMI, β-CTX, duration of GC use, BMD, and BA were independent impact factors of height growth. The relationship between obesity and the side effects of GC has been studied and formally monitored in the management of DMD. In DMD, this risk of obesity vulnerability is increased by GC, reduced mobility, and limited opportunities for physical activity resulting in reduced energy expenditure [24]. Gluco-corticoids have multiple side effects such as obesity, growth suppression, and impaired bone health, and to some extent, these adverse effects are reflected in height development. The inter-action between these factors is complex, and long-term GC exposure usually exacerbates endo-crine dysfunction in the organism.

Our research proposes, for the first time, the potential value of altered trends in bone mark-ers for growth and development in DMD. β-CTX is a marker of bone resorption, which reflects the degree of bone matrix degradation [25]. Bone resorption gains with age in healthy populations suggest a promotion of new bone formation, as bone resorption is a source of bone formation-stimulating factors [26,27]. Regarding children's development, the β-CTX expression profile differs from other bone turnover markers such as PINP, as it does not show a peak in early infancy. Existing studies have found that, throughout childhood, β-CTX expres-sion remains relatively stable from birth to 12 years old, with a slight increase leading up to early puberty, followed by a decrease [28]. In our investigation of DMD boys, β-CTX is posi-tively related to height, which is quite congruent with the positive correlations between height and β-CTX in healthy children from 2 months to 18 years [29]. Moreover, prior exploration had shown, a gain in bone resorption with age and reached the peak during Tanner stage 4 (A level of classification in the 5-stage categorical sexual development level evaluation method by Tanner) in healthy males [29,30]. However, the plasma level of β-CTX experienced a down-ward trend after the onset of GC therapy in our DMD cohort. The use of GC may have exacer-bated the effect of β-CTX on height, the duration of which has been shown in our model to be directly proportional to the rate of decrease in β-CTX levels. The decrease in β-CTX levels dur-ing the first year of GC initiation suggests that GC may indirectly affect height through bone metabolism. Thus, regular monitoring of bone turnover markers can be effective in managing bone health in DMD.

We find that the portion of short stature was significantly raised during the first year of GC use, and relevant bone health markers showed different sequential changing patterns. As GC use is prolonged, the level of β-CTX initially decreases, which is a sensitive indicator of affected bone health, after which the proportion of delayed BA gradually increases. Then, the bone mineral density shows a decreasing trend only after the third year of GC use, which reflects the long-term effect of GC exposure on bone health. Specifically, the rate of low BMD fell during the first two years of GC use, but then rose sharply in the third year, far exceeding the previous portion. It has long been shown that in healthy individuals, peak bone mass is almost completely reached by late adolescence or early adulthood [31]. However, our result revealed that BMD kept accumulating during the first two years of prolonged GC exposure (aged around 5.5 to 7.5), yet after which it greatly slumped, probably influenced by both progressive myopathy and cumulative osteo-toxicity of steroids [32]. This was in agreement with the previ-ous findings of our team, where we have reported that children with DMD no longer maintain their BMD leave at least age 8 years [33]. Additionally, the growth of bone mineral content lags

behind the growth of height [34], and the dissociation between bone expansion and bone mineralization during the growing period may result in relative skeletal fragility [35], which may provide a further explanation for the heterogeneity in the development of stature and BMD in the DMD population.

In addition, we found an almost consistent trend for a decrease in delayed BA and short stature after GC use. Bone age is a quantitative measurement of skeletal maturation [36], when researchers corrected the height SDS by BA, the number of participants with short stature was reduced from 52.5% to 22.0%, which suggested that the stunted growth was reflected in delayed skeletal maturation [37]. As an apparent influence factor on stature and puberty, the interaction between BA, BMD, bone markers, and other potential factors related to bone health needs to be further investigated. We suggest shedding light on the height growth and focus on those sensitive biomarkers that tend to aggravate in the first year of GC exposure to predict possible changes in bone health conditions.

Our study innovated to present the effect of GC, bone turnover marker, BMD, and BA on the stature of DMD and figure out the key point time of variations. Bone turnover markers have predictive value in height for patients with DMD, especially β-CTX. However, the prediction of a single biomarker may not be precise enough, and further in-depth studies are needed to support their use as a reliable basis for clinical prediction and decision.

This study provides evidence for the early management of height development to manage future bone health of DMD. The occurrence and progression of short stature are unequivocally associated with patients' broad bone health status, with bone turnover markers, BMD, and BA as independent indicators. We recommend a comprehensive option containing these monitoring items to guide clinical follow-up management and treatment. Different bone indicators have different sensitivities and specificities and need to be considered together for clinical monitoring of bone health. Also, it is essential to monitor height during the early stages of the disease while focusing on bone health to minimize the occurrence of bone-related complications later in life. Monitoring of height and relative impactors should become a routine of DMD early care to help optimize treatments to reduce delay in skeletal maturation, contribute to optimal catch-up growth, and maintain bone health.

Still, this study has some limitations. Firstly, it only includes patients aged 2–12, excluding adolescent and adult patients. To gain a better understanding of the global growth pattern of DMD, particularly during puberty and the non-ambulatory stage, a wider age range of participants needs to be involved. Secondly, the children who were followed up by our team were examined for spinal health (including scoliosis and spina bifida/fracture) every six months to one year. The participants in this cohort were all under age 12, most of them have healthy spine and only few of them have very mild curve that is lower than 10° (which did not meet the diagnostic criteria of scoliosis that cobb > 10°), so scoliosis, increased lumbar lordosis and contractures were not assessing. A mild curvature of spine can also affect the accuracy of height measurement, although height measurement for scoliosis is difficult, more comprehensive approaches should be tried to improve the accuracy of height measurement, including the use of X-ray assisted measurement. Additionally, a longitudinal, multi-center study should be conducted to minimize bias.

## Acknowledgments

The authors wish to acknowledge Dr. Sophelia Chan, Professor of the Department of Paediatrics and Adolescent Medicine at the School of Clinical Medicine, The University of Hong Kong, for her assistance in interpreting the results of this study.

## Author Contributions

**Conceptualization:** Xiaotang Cai, Qiu Wang, Xiaomei Sun.

**Data curation:** Linyuhan Zhou, Shuangru Li, Qin Hu, Min Huang, Dan Zhou.

**Formal analysis:** Bingying Wang, Linyuhan Zhou, Shuangru Li.

**Funding acquisition:** Huayan Xu, Xiaotang Cai.

**Investigation:** Bingying Wang, Linyuhan Zhou, Shuangru Li, Qin Hu, Min Huang.

**Methodology:** Bingying Wang, Linyuhan Zhou, Shuangru Li, Huayan Xu, Yingkun Guo, Dan Zhou, Qiu Wang.

**Project administration:** Xiaotang Cai.

**Resources:** Huayan Xu, Yingkun Guo, Qin Hu, Min Huang, Dan Zhou, Xiaomei Sun.

**Software:** Bingying Wang, Linyuhan Zhou, Shuangru Li.

**Supervision:** Xiaotang Cai, Qiu Wang, Xiaomei Sun.

**Validation:** Qiu Wang, Xiaomei Sun.

**Visualization:** Bingying Wang.

**Writing – original draft:** Bingying Wang, Linyuhan Zhou, Shuangru Li.

**Writing – review & editing:** Xiaotang Cai, Qiu Wang, Xiaomei Sun.

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
