## [Decision Letter · Decision Letter 0]

25 Sep 2024

PONE-D-24-27979Height development and multiple bone health indicators in children aged 2-12 years with Duchenne muscular dystrophy (DMD)PLOS ONE

Dear Dr. Cai,

Thank you for submitting your manuscript to PLOS ONE. After careful consideration, we feel that it has merit but does not fully meet PLOS ONE’s publication criteria as it currently stands. Therefore, we invite you to submit a revised version of the manuscript that addresses the points raised during the review process.

We look forward to receiving your revised manuscript.

Kind regards,

Claudia Brogna

Academic Editor

PLOS ONE

**Journal requirements:**

2. Please provide additional details regarding participant consent. In the ethics statement in the Methods and online submission information, please ensure that you have specified (a) whether consent was informed and (b) what type you obtained (for instance, written or verbal, and if verbal, how it was documented and witnessed). If your study included minors, state whether you obtained consent from parents or guardians. If the need for consent was waived by the ethics committee, please include this information.

This work was supported by grants from the Sichuan Science and Technology Support Program :2023YFG0284 & 2023ZYD0121 (receiver:Xiaotang Cai), Natural Science Foundation of Sichuan Province :24NSFC1085 (receiver:Huayan Xu), and the Chengdu Municipal Health Commission Project: 21PJ048 (receiver: Xiaotang Cai).

URL:

Sichuan Science and Technology Support & Natural Science Foundation of Sichuan

Province: https://kjt.sc.gov.cn/ Program

Chengdu Municipal Health Commission Project：https://cdwjw.chengdu.gov.cn/

There is not any sponsors or funders play any role in this study.

5. In this instance it seems there may be acceptable restrictions in place that prevent the public sharing of your minimal data. However, in line with our goal of ensuring long-term data availability to all interested researchers, PLOS’ Data Policy states that authors cannot be the sole named individuals responsible for ensuring data access (http://journals.plos.org/plosone/s/data-availability#loc-acceptable-data-sharing-methods).

Reviewers' comments:

Reviewer's Responses to Questions

**Comments to the Author**

1. Is the manuscript technically sound, and do the data support the conclusions?

Reviewer #1: Partly

Reviewer #2: Yes

2. Has the statistical analysis been performed appropriately and rigorously? 

Reviewer #1: Yes

Reviewer #2: Yes

3. Have the authors made all data underlying the findings in their manuscript fully available?

Reviewer #1: Yes

Reviewer #2: Yes

4. Is the manuscript presented in an intelligible fashion and written in standard English?

Reviewer #1: Yes

Reviewer #2: Yes

5. Review Comments to the Author

**Reviewer #1:** Bone health impairment and short stature are significant issues faced by children with Duchenne Muscular Dystrophy (DMD), regardless of whether they are undergoing glucocorticoid treatment. This manuscript is valuable for its aim to shed light on these problems. However, there are several critical points that require clarification:

1. The authors mention that height and weight were measured using standard equipment. It would be beneficial to clarify what is meant by "standard equipment." Additionally, in ambulatory patients, accurate height measurement can be challenging in DMD children due to factors such as scoliosis, increased lumbar lordosis, and contractures. Therefore, it is important to specify the percentage of patients with scoliosis, contractures, and increased lumbar lordosis. For non-ambulatory patients, it is evident that height measurement cannot be performed using standard equipment, so a detailed explanation is necessary.

2. The abbreviation "MDT" in line 114 should be explained.

3. Bone Mineral Density (BMD) is typically expressed in units of g/cm². The authors should clarify their use of the mg/cc unit.

4. The authors report a vertebral fracture (VF) rate of 0% in their patients. However, asymptomatic vertebral fractures can occur in 30-50% of DMD children, especially those on glucocorticoid therapy. Therefore, annual (or more frequent if symptoms are present) monitoring with lateral spinal radiographs is recommended. The manuscript does not clearly state the frequency of routine follow-ups, nor whether asymptomatic vertebral fractures were actively investigated through lateral spinal radiographs. Without this monitoring, claiming a 0% VF rate may not be accurate.

5. Are there any patients with long bone fractures, diagnosed osteoporosis, or those receiving bisphosphonate treatment?

6. In Figure 1-d, the authors should explain why height growth velocity is so low at age 3 in the DMD population.

**Reviewer #2:** The authors present an impressive large cohort of early- to late- ambulatory DMD patients and their associated growth parameters and assessments of bone health.

Introduction

Givinostat now also shown to be effective so glucocorticoids are no longer the only pharmacologic option.

Line 82: What is "Molly"?

Methods

For how many subjects was parents' stature known re. the exclusion criterion? This may be documented in Endocrine assessments but isn't typical in neurology clinic.

Was DXA and bone age measured at the same age in all participants? How many had multiple available?

Results

Table 1 and 2 require a legend explaining the abbreviations

Figure one: why is the data point for 9 years in D so low?

Line 153 - "stature" to replace "statue"

Line 170 - "affecting" to replace "affect"

Discussion

Recommend combining the bone marker discussion into one rather than repeating it - the first discussion of its use in clinical care is overly strong given lack of normative pediatric data and variability; the discussion is more fairly balanced when repeated later on.

Comment on the limitation of not assessing contractures or scoliosis in the data which both impact height and BMI measurement.

Line 273: Unclear sentence: Earlier study rather than age? After the age of 8 years?

Line 302: suggest "adult" as "elderly" is not typical of the DMD population.

This data is remarkable for the size of the cohort and will be useful to clinicians in understanding the trajectory of bone health in early stages of DMD.

6. PLOS authors have the option to publish the peer review history of their article (what does this mean?). If published, this will include your full peer review and any attached files.

Reviewer #1: **Yes: **Dr. Meral Bilgilisoy Filiz

Reviewer #2: No

---

## [Author Response · Author response to Decision Letter 0]

4 Nov 2024

Dear editors and reviewers,

Thank you for the detailed review and the precious opportunity to revise our manuscript titled “Height development and multiple bone health indicators in children aged 2-12 years with Duchenne muscular dystrophy (DMD)” (ID: PONE-D-24-27979). We are very grateful for your constructive comments and suggestions, which are very valuable and helpful for improving our manuscript. In the following, the responses to all the comments are provided one by one.

We have tried our best to make all the revisions clear, and we hope that the revised manuscript can satisfy the requirements for publication.

The main revisions in the new manuscript are:

1.Line 42 - Edited funding resource.

2.Line 77 - Added the abbreviation of MDT.

3.Line 108 - “Lamb et al” to replace “Molly”

4.Line 134&136 - Details of the ethics approval have been reported.

5.Line 147 - Frequency of regular monitoring has been supplemented

6.Line 152 - Added the full name of “MDT”, which stands for “Multidisciplinary Team”.

7.Line 162~167 - Added detailed methods for height and weight measurement.

8.Line 206 - Added the explanations for the abbreviations of table 1.

9.Line 211 - “stature” to replace “statue”

10.Line 235 - “affecting” to replace “affect”

11.Line 252 - Added the meaning of symbol “*”

12.Line 253 - Added the explanations for the abbreviations of table 2.

13.Line 348 - The description of bone turnover markers has been revised .

14.Line 380 - the unclear sentence had been revised.

15.Line 429 - “adult” to replace “elderly”.

16.Line 432 - The limitation of measuring method was added.

Responses to reviewer 1#

1.The authors mention that height and weight were measured using standard equipment. It would be beneficial to clarify what is meant by "standard equipment." Additionally, in ambulatory patients, accurate height measurement can be challenging in DMD children due to factors such as scoliosis, increased lumbar lordosis, and contractures. Therefore, it is important to specify the percentage of patients with scoliosis, contractures, and increased lumbar lordosis. For non-ambulatory patients, it is evident that height measurement cannot be performed using standard equipment, so a detailed explanation is necessary. 

Response:

Thank you for this constructive suggestion. There is some inevitable bias, but we can confirm that its impact on the accuracy of the measurements is minimal by the quality control.

In this study, a calibrated device was used to measure height and weight. Each individual was measured three times, and the average value was recorded for accuracy. Also, the children were examined for spinal health (including scoliosis and spina bifida/fracture) every six months. Because the participants in this study were all under 12 years old, most of them have healthy spine and only few of them have very mild curve that is lower than 10° (which did not meet the diagnostic criteria of scoliosis that cobb > 10°) and have little impact on measurement accuracy. Moreover, for participants with scoliosis (cobb >10°), severe contractures, or inability to stand independently, the ulnar calculation and are span have been performed for height. The exact formula and reference are as follows:

Calculation formula: height (cm) = (4.605* ulna length)+(1.308*age in years)+28.003

Reference: 1) Gauld LM, et al. Height prediction from ulna length. Dev Med Child Neurol. 2004 Jul;46(7):475-80. 2) Forman MR, et al. Arm span and ulnar length are reliable and accurate estimates of recumbent length and height in a multiethnic population of infants and children under 6 years of age. J Nutr. 2014 Sep;144(9):1480-7.

According to the reviewer’s insight advice, we have added description of our measuring methods in the manuscript, and limitation of height measurement for children with minor curvature of the spine (which is not clinically significant) had been added, we will try more comprehensive methods to reduce measurement errors.

2.The abbreviation "MDT" in line 114 should be explained. 

Response：

Thank you for your kind suggestion. explanation for “MDT” has been added, which stands for “Multidisciplinary Team”.

3.Bone Mineral Density (BMD) is typically expressed in units of g/cm². The authors should clarify their use of the mg/cc unit. 

Response: 

Thank you for the suggestion. g/cm² is the unit for the DAX measurement, which measures flat data for standard screening, while mg/cc is the unit for the QCT measurement which provides 3D imaging data for detailed bone architecture analysis. Our BMD data were provided by QCT.

4.The authors report a vertebral fracture (VF) rate of 0% in their patients. However, asymptomatic vertebral fractures can occur in 30-50% of DMD children, especially those on glucocorticoid therapy. Therefore, annual (or more frequent if symptoms are present) monitoring with lateral spinal radiographs is recommended. The manuscript does not clearly state the frequency of routine follow-ups, nor whether asymptomatic vertebral fractures were actively investigated through lateral spinal radiographs. Without this monitoring, claiming a 0% VF rate may not be accurate. 

Response: 

Thank you for the suggestion. The frequency of routine monitoring had been added in our manuscript. 

According to other research, the prevalent of VF in DMD patients aged 7-12 with daily oral prednisone is about 2% from UK NorthStar database. For our study, the participants were aged 2-12, which is also too young to have high VF rate, while the regular monitoring (spinal radiographs every 6 to 12 months), standarized care of our multidisciplinary team (pediatric neurology, endocrinology, radiology, cardiovascular, rehabilitation, nutrition), early intervention, and tailored therapies (including spinal management) helped children with DMD maintain muscle strength and bone health. In our center, the VF prevalent was about 2%, consistent with the prevalence in other researches.

5.Are there any patients with long bone fractures, diagnosed osteoporosis, or those receiving bisphosphonate treatment? 

Response: 

Thank you for the suggestion. 8 participants had a previous fracture, including ankle, elbow, femur, upper arm, tibia and nose, most of these fractures happened in their early age before GC use. 91 participants with BMD z-score < -2 were diagnosed osteoporosis. 6 participants with oral Alendronate sodium and 7 participants with zoledronic acid injection in this study. 

6. In Figure 1-d, the authors should explain why height growth velocity is so low at age 3 in the DMD population. 

Response: 

Thank you for this question. As diagnosis of DMD is usually after 3 years old, the sample size of age 3 was relatively small which may lead to bias in results. With improvement of people’s awareness and diagnostic methods, the age of diagnosis is gradually advanced, more data before age 3 will be included in our cohort.

Responses to reviewer 2#

Line 82: What is "Molly"?

Response: 

We apologize that we confused the first and last name of this researcher, thank you for finding out. We had corrected this in the manuscript. 

For how many subjects was parents' stature known re. the exclusion criterion? This may be documented in Endocrine assessments but isn't typical in neurology clinic.

Response: 

Just as you mentioned, there are Endocrinologists in the multidisciplinary team to conduct the growth and stature monitoring. At the time of writing, no patient at our center meets this exclusion criterion in question. Nevertheless, this was retained in the manuscript because, in endocrinology, very short stature in adult parents may be regarded as a form of short stature that deviates from the familial genetic background.

Was DXA and bone age measured at the same age in all participants? How many had multiple available? 

Response: 

Thank you for the questions. Patients were followed-up in our center every 6 month to one year. BMD (QCT) was not measured each visit, in accordance with guideline, we started to measure BMD around the onset of GC use, BMD was measured at least once a year. Endocrinologists will conduct bone age testing according to children's growth and development needs. In this study, all participants were measured at the same age, 291 multiple data (BMD and bone age) were available.

Results

Table 1 and 2 require a legend explaining the abbreviations

Repones: 

Thank you for your suggestion, explanation for abbreviations had been added.

Figure one: why is the data point for 9 years in D so low? 

Response: 

Figure1-D shows the growth velocity, it indicates that children's height is still increasing but at a slower rate by the age 9. The total sample size is large, but the sample size in group age 9 might insufficient, it can lead abnormal distribution, instability and bias in the statistical results. As the number of patients in our cohort increases and more repeated-measures data are collected, a longitudinal study with a larger simple supplement and verify the results in our future study. 

Line 153 - "stature" to replace "statue"

Line 170 - "affecting" to replace "affect"

Line 302: suggest "adult" as "elderly" is not typical of the DMD population. 

Reponse: 

Thank you for all the detailed corrections. we replace these words for more accurate statements

Discussion

Recommend combining the bone marker discussion into one rather than repeating it - the first discussion of its use in clinical care is overly strong given lack of normative pediatric data and variability; the discussion is more fairly balanced when repeated later on.

Response: 

Thank you for the suggestion. This part has been condensed in the manuscript.

Comment on the limitation of not assessing contractures or scoliosis in the data which both impact height and BMI measurement. 

Repones: 

Thank you for your helpful advice. We have added a discussion on the impact of scoliosis on height measurement in limitation section. 

The children who were followed up by our team were examined for spinal health (including scoliosis and spina bifida/fracture) every six months to one year. The participants in this cohort were under age 12, most of them did not meet the diagnostic criteria of scoliosis (cobb > 10°). A mild curvature of spine can also affect the accuracy of height measurement, although height measurement for scoliosis is difficult, we will apply more comprehensive approach to improve the accuracy of height measurement, including the use of X-ray assisted measurement.

Line 273: Unclear sentence: Earlier study rather than age? After the age of 8 years? 

Response:

We apologize for the unclear expression. We had revised the statements to make the sentences clearer. The revised paragraph is as follows:

It has long been shown that in healthy individuals, peak bone mass is almost completely reached by late adolescence or early adulthood[30]. However, our result revealed that BMD kept accumulating during the first two years of prolonged GC exposure (aged around 5.5 to 7.5), yet after which it greatly slumped, probably influenced by both progressive myopathy and cumulative osteo-toxicity of steroids[31]. This was in agreement with the previous findings of our team, where we have reported that children with DMD no longer maintain their BMD leave at least age 8 years[32].

PLOS authors have the option to publish the peer review history of their article (what does this mean?). If published, this will include your full peer review and any attached files.

Response:

Yes, we agree to publish the peer review history of this article.

Thanks to the professional comments again that point out the above problems. The manuscript has revised carefully and thoroughly according to each point. We hope these explanations would answer the doubts.

Sincerely,

Dr. Xiaotang Cai

---

## [Decision Letter · Decision Letter 1]

1 Dec 2024

PONE-D-24-27979R1Height development and multiple bone health indicators in children aged 2-12 years with Duchenne muscular dystrophy (DMD)PLOS ONE

Dear Dr. Cai,

Thank you for submitting your manuscript to PLOS ONE. After careful consideration, we feel that it has merit but does not fully meet PLOS ONE’s publication criteria as it currently stands. Therefore, we invite you to submit a revised version of the manuscript that addresses the points raised during the review process.

Please submit your revised manuscript by Jan 15 2025 11:59PM. If you will need more time than this to complete your revisions, please reply to this message or contact the journal office at plosone@plos.org. Please include the following items when submitting your revised manuscript:A rebuttal letter that responds to each point raised by the academic editor and reviewer(s). You should upload this letter as a separate file labeled 'Response to Reviewers'.A marked-up copy of your manuscript that highlights changes made to the original version. You should upload this as a separate file labeled 'Revised Manuscript with Track Changes'.An unmarked version of your revised paper without tracked changes. You should upload this as a separate file labeled 'Manuscript'.If applicable, we recommend that you deposit your laboratory protocols in protocols.io to enhance the reproducibility of your results. Protocols.io assigns your protocol its own identifier (DOI) so that it can be cited independently in the future. For instructions see: https://journals.plos.org/plosone/s/submission-guidelines#loc-laboratory-protocols. Additionally, PLOS ONE offers an option for publishing peer-reviewed Lab Protocol articles, which describe protocols hosted on protocols.io. Read more information on sharing protocols at https://plos.org/protocols?utm_medium=editorial-email&utm_source=authorletters&utm_campaign=protocols.

We look forward to receiving your revised manuscript.

Kind regards,

Claudia Brogna

Academic Editor

PLOS ONE

Journal Requirements:

Reviewers' comments:

Reviewer's Responses to Questions

**Comments to the Author**

1. If the authors have adequately addressed your comments raised in a previous round of review and you feel that this manuscript is now acceptable for publication, you may indicate that here to bypass the “Comments to the Author” section, enter your conflict of interest statement in the “Confidential to Editor” section, and submit your "Accept" recommendation.

Reviewer #1: All comments have been addressed

Reviewer #2: All comments have been addressed

2. Is the manuscript technically sound, and do the data support the conclusions?

Reviewer #1: (No Response)

Reviewer #2: Yes

3. Has the statistical analysis been performed appropriately and rigorously? 

Reviewer #1: (No Response)

Reviewer #2: Yes

4. Have the authors made all data underlying the findings in their manuscript fully available?

Reviewer #1: (No Response)

Reviewer #2: Yes

5. Is the manuscript presented in an intelligible fashion and written in standard English?

Reviewer #1: (No Response)

Reviewer #2: No

6. Review Comments to the Author

Reviewer #1: (No Response)

Reviewer #2: Recommend an additional revision of the language for grammatical fluency.

Two small edits:

"Tanner' is misspelled in second use line 356.

The sentence starting in 376 is repeated further down the page, with a difference reference - one should be deleted.

7. PLOS authors have the option to publish the peer review history of their article (what does this mean?). If published, this will include your full peer review and any attached files.

Reviewer #1: **Yes: **Meral Bilgilisoy Filiz

Reviewer #2: No

---

## [Author Response · Author response to Decision Letter 1]

3 Dec 2024

Response Letter

Dear editors and reviewers,

Thank you for the detailed review and the precious opportunity to revise our manuscript titled “Height development and multiple bone health indicators in children aged 2-12 years with Duchenne muscular dystrophy (DMD)” (ID: PONE-D-24-27979). We are very grateful for your constructive comments and suggestions, which are very valuable and helpful for improving our manuscript. In the following, the responses to all the comments are provided one by one.

We have tried our best to make all the revisions clear, and we hope that the revised manuscript can satisfy the requirements for publication.

The main revisions in the new manuscript are:

1.Line 358 - “Tanner” to replace “Tunner”

2.Line 381 - Repeated sentence and reference had been deleted.

Responses to reviewer 2#

"Tanner' is misspelled in second use line 356.

Response:

We appreciate your careful reading of our manuscript. The misspelled word have been corrected, thank you for finding out. 

The sentence starting in 376 is repeated further down the page, with a difference reference - one should be deleted.

Response:

We sincerely apologize for the oversight of repeating a sentence in the manuscript. We have removed the duplicate sentence in the revised version. Thank you for your attention to detail, which has been invaluable in improving the clarity and quality of our work.

Thank you once again to the professional comments that point out the above problems. The manuscript has revised carefully and thoroughly according to each point. We hope these explanations would answer the doubts.

Sincerely,

Dr. Xiaotang Cai

---

## [Editor Report · Decision Letter 2]

18 Dec 2024

Height development and multiple bone health indicators in children aged 2-12 years with Duchenne muscular dystrophy (DMD)

PONE-D-24-27979R2

Dear Dr. Xiaotang Cai, 

We’re pleased to inform you that your manuscript has been judged scientifically suitable for publication and will be formally accepted for publication once it meets all outstanding technical requirements.

Kind regards,

Claudia Brogna

Academic Editor

PLOS ONE
---

## [Editor Report · Acceptance letter]

2 Jan 2025

PONE-D-24-27979R2 

PLOS ONE

Dear Dr. Cai, 

I'm pleased to inform you that your manuscript has been deemed suitable for publication in PLOS ONE. Congratulations! Your manuscript is now being handed over to our production team.

Kind regards, 

on behalf of

Dr. Claudia Brogna 

Academic Editor

PLOS ONE